# Piezoelectric Characteristics of 0.55Pb(Ni_1/3_Nb_2/3_)O_3_-0.45Pb(Zr,Ti)O_3_ Ceramics with Different MnO_2_ Concentrations for Ultrasound Transducer Applications

**DOI:** 10.3390/ma12244115

**Published:** 2019-12-09

**Authors:** Myeongcheol Kang, Lae-Hyong Kang

**Affiliations:** 1Department of Mechatronics Engineering and LANL-JBNU Engineering Institute Korea, Jeonbuk National University, 567 Baekje-daero, Deokjin-gu, Jeonju-si, Jeollabuk-do 54896, Korea; mckang@jbnu.ac.kr; 2Department of Flexible and Printable Electronics, Department of Mechatronics Engineering, and LANL-JBNU Engineering Institute Korea, Jeonbuk National University, 567 Baekje-daero, Deokjin-gu, Jeonju-si, Jeollabuk-do 54896, Korea

**Keywords:** piezoelectric ceramic, 0.55Pb(Ni_1/3_Nb_2/3_)O_3_-0.45Pb(Zr,Ti)O_3_, MnO_2_ doping, electrical properties

## Abstract

In this study, we investigate the piezoelectric characteristics of 0.55Pb(Ni_1/3_Nb_2/3_)O_3_-0.45Pb(Zr,Ti)O_3_ (PNN-PZT) with MnO_2_ additive (0, 0.25, 0.5, 1, 2, and 3 mol%). We focus on the fabrication of a piezoelectric ceramic for use as both actuator and sensor for ultrasound transducers. The actuator and sensor properties of a piezoelectric ceramic depend on the piezoelectric strain coefficient *d* and piezoelectric voltage coefficient *g*, related as *g* = *d*/*ε*^T^. To increase *g*, the dielectric constant *ε*^T^ must be decreased. PNN-PZT with MnO_2_ doping is synthesized using the conventional solid-state reaction method. The electrical properties are determined based on the resonant frequencies and vibration modes measured by using an impedance analyzer. The MnO_2_ addition initially improves the tetragonality of the PNN-PZT ceramic, which then saturates at a MnO_2_ content of 1 mol%. Therefore, the dielectric constant and piezoelectric coefficient *d*_33_ steadily decrease, while the mechanical properties (*Q*_m_, Young’s modulus), tan*δ*, electromechanical coupling coefficient *k*, and piezoelectric voltage coefficient *g* were improved at 0.5–1 mol% MnO_2_ content.

## 1. Introduction

Piezoelectric materials have attracted considerable interest for various applications such as multi-layer ceramic actuator, transducer, sensor and actuator applications, and for analyses on fundamental science. Lead-based piezoelectric ceramics such as Pb(Zr,Ti)O_3_ (PZT) have been extensively used in electrical devices because of their excellent piezoelectric properties [1]. Recently, the policies suggesting lead elimination have triggered studies on alternative compounds such as (K _0.44_Na_0.52_Li_0.04_)-(Nb_0.86_Ta_0.10_Sb_0.04_)O_3_ (KNL-NTS) [2]. However, the lead-based compositions have exhibited higher piezoelectric performances than those of the lead-free compositions such as KNL-NTS.

Recently, many piezoelectric ceramic compositions of the ternary system have been reported as the ternary system has a larger morphotropic phase boundary (MPB) area than that of a secondary system such as PZT. Lead-based relaxor ferroelectrics have a general formula of Pb(B’ B”)O_3_ (where B’ is Mg^2+^, Ni^2+^, Zn^2+^, Fe^3+^, Sc^3+^, In^3+^, Mn^4+^, and Sn^4+^ and B” is Nb^5+^, Ta^5+^, Sb^5+^, and W^6+^) [3,4]. In this ternary system, 0.55Pb(Ni_1/3_Nb_2/3_)O_3_-0.45Pb(Zr,Ti)O_3_ (PNN-PZT) has attracted increasing attention owing to its excellent piezoelectric properties. Modifications of lead-based relaxor ferroelectrics have been extensively investigated to improve the piezoelectric properties. In 1974, Luff et al. [5] investigated a 0.5Pb(Ni_1/3_Nb_2/3_)O_3_-0.35PbTiO_3_-0.15PbZrO_3_ solid solution in the PNN-PZT ternary system and observed excellent piezoelectric properties. Vittayakorn et al. [6] reported the phase diagram between PNN and PZT ((1–*x*)Pb(Ni_1/3_Nb_2/3_)O_3_-*x*Pb(Zr_1/2_Ti_1/2_)O_3_ (*x* = 0.4–0.9)), where the Zr/Ti composition was fixed close to the MPB of PZT, and new MPB within this system. Cao et al. [7] investigated 0.3Pb(Ni_1/3_Nb_2/3_)O_3_-*x*PbTiO_3_-(0.7−*x*)PbZrO_3_ (*x* = 0.33–0.43) as a function of the content of PbTiO_3_. Nam et al. [8] reported the 0.35Pb(Ni_1/3_Nb_2/3_)O_3_-0.65Pb(Zr_1−*x*_Ti*_x_*)O_3_ (*x* = 0.56–0.63) ceramic composition sintered at 1200 °C with good electrical properties including *d*_33_ = 605 pC/N, *K*_p_ = 0.61, and *ε*_r_ = 3600. Du et al. [3] have reported the 0.55Pb(Ni_1/3_Nb_2/3_)O_3_-0.135PbZrO_3_-0.315PbTiO_3_ ceramic with good electrical properties including *d*_33_ = 1070 pC/N, *K*_p_ = 0.69, *ε*_r_ = 8710, and tan*δ* = 26 × 10^−3^. Figure 1 shows the reported compositions of the PNN-PZT ceramics.

The doping of proper elements is an effective approach to enhance their properties for targeted applications. Yu et al. [9] investigated the effects of MnO_2_ additive on the 0.12Pb(Ni_1/3_Sb_2/3_)-0.48PbTiO_3_-0.40PbZrO_3_ ceramic. A 0.15 wt% MnO_2_-doped sample exhibited enhanced piezoelectric properties, including *K*_p_ = 0.68, *ε*_r_ = 3069, *Q*_m_ = 181, and tan*δ* = 5.4 × 10^−3^. Bamiere et al. [10] reported that a Sr-doped 0.674Pb,Nd(Zr,Ti)O_3_-Pb(Ni_1/3_Nb_2/3_)O_3_ exhibited good ferroelectric properties at low sintering temperatures. Du et al. [11] investigated the effects of small additions of *x*Fe_2_O_3_ (*x* = 0~1.6 mol%) on the microstructures and electrical properties of 0.55Pb(Ni_1/3_Nb_2/3_)O_3_-0.45Pb(Zr_0.3_Ti_0.7_)O_3_ ceramics sintered at 1200 °C for 2 h and reported high piezoelectric performances at 1.2 mol% of Fe_2_O_3_ (*ρ* = 7.97 g/cm^3^, *d*_33_ = 956 pC/N, *K*_p_ = 0.74, *ε*_r_ = 6095, and tan*δ* = 26 × 10^−3^). Yoo et al. [12] reported high values of piezoelectric properties (*ρ* = 7.816 g/cm^3^, *d*_33_ = 356 pC/N, *k*_p_ = 0.597, *ε*_r_ = 920, *Q*_m_ = 1186) for the 0.2 wt% MnO_2_ 0.02Pb(Mn_1/3_Nb_2/3_)O_3_-0.12Pb(Ni_1/3_Nb_2/3_)O_3_-0.86Pb(Zr_0.5_Ti_0.5_)O_3_ ceramic. Liao et al. [13] reported that Fe doping largely reduced tan*δ* and *d*_33_, but improved *Q*_m_ and tetragonality of the 0.35BiScO_3_-0.6PbTiO_3_-0.05Pb(Zn_1/3_Nb_2/3_)O_3_ ceramic. Liu et al. [4] investigated 0.55Pb(Ni_1/3_Nb_2/3_)O_3_-0.45Pb(Zr_0.3_Ti_0.7_)O_3_ (PNN-PZT) with varying MnO_2_ additive content and reported high performance (*d*_33_ = 710 pC/N, *k*_p_ = 0.595, *ε*_r_ = 3092.25, tan*δ* = 14.9 × 10^−3^, *Q*_m_ = 176) at a MnO_2_ content of 1 mol%. Table 1 shows piezoelectric ceramic and properties according to doping from references.

Most lead-based relaxor ferroelectrics such as PNN-PZT have high dielectric constants and *d*_33_ values, but low *g*_33_ values and poor mechanical properties (*Q*_m_, Young’s modulus) for applications as sensors [13]. Modifications of the properties of the ternary complex systems have not been extensively investigated. The doping of materials such as FeO_3_ and MnO_2_ can be used to easily change the properties of ternary complex systems. To improve the *g*_33_ values and mechanical properties of the PNN-PZT systems, Mn can be added to selectively improve sensor characteristics. The MnO_2_ doping can improve the sinterabilities of lead-based relaxor ceramics owing to the increase in number of oxygen vacancies, generated by the substitutions of the high-valence Ti^4+^ and Zr^4+^ in the perovskite lattice by the low-valence Mn^2+^ and/or Mn^3+^ [4]. The oxygen vacancies in the perovskite lattice can promote lattice diffusion, thus assisting the sintering and grain growth [12], as the grain boundary movement is dragged by these defects. The pores in the lead-based relaxor ceramics easily diffuse through the movement of oxygen vacancies and are eliminated at the grain boundaries. Therefore, the densification of the ceramic originated from the introduction of MnO_2_ improving the piezoelectric properties [13]. It is possible to adjust the properties of the PNN-PZT ceramic according to the MnO_2_ content.

The aim of this study was to better understand the effects of doping on PZT-based complex ceramics and improve the sensing performances and mechanical properties by using MnO_2_ as an additive in the 0.55Pb(Ni_1/3_Nb_2/3_)O_3_-0.135PbZrO_3_-0.315PbTiO_3_ ternary ceramic. The effects of the MnO_2_ content on the piezoelectric, dielectric, and mechanical properties of the ceramic were investigated.

## 2. Synthesis of the MnO_2_-Doped PNN-PZT

The ceramics of PNN-PZT + MnO_2_ (0, 0.25, 0.5, 1, 2, and 3 mol%) were synthesized by using the conventional solid-state method. Raw material powders (PbO, NiO, Nb_2_O_5_, ZrO_2_, and TiO_2_) (Sigma Aldrich, 99.99%, Gillingham, UK) and MnO_2_ additive (Sigma Aldrich, 99.99%, Gillingham, UK) were weighted in chemically stoichiometric proportions and ball-milled with distilled water for 24 h. After the ball milling, the slurry was dried at 80 °C, and then calcined at 900 °C (heating rate: 100 °C/h) for 2 h. The calcined powders were ball-milled again for 24 h with a 5 wt% polyvinyl alcohol (PVA) (Sigma Aldrich, 99+%, Gillingham, UK, *M*_w_ = 89000–98000) solution as a binder for ceramic formation. The mixture was dried and crushed by a high-energy ball mill. The powder was sieved to control the particle size below 5 μm. The sieved powder was pressed into a mold (Φ = 30 mm) under a pressure of 100 MPa. The PVA in the ceramic disk was burnt out at 600 °C (heating rate: 100 °C/h) for 2 h. Subsequently, the ceramic disk was sintered at 1200 °C (heating rate: 100 °C/h) for 2 h with a spacer powder in an alumina crucible [14]. The sintered PNN-PZT ceramic disk was polished and coated with silver paste to obtain electrodes on both surfaces. The samples were polarized by applying a direct-current (DC) electric field of 2.5 kV/mm for 30 min at 50 °C in silicon oil.

The crystal structures of the sintered samples were characterized by X-ray diffraction (XRD) (X’pert Pro Powder, PANalytical, Netherlands). The surface microstructures of the as-sintered ceramics were observed by using field-emission scanning electron microscopy (SEM) (JSM-5900, JEOL, Akishima City, Japan). Three PNN-PZT specimens were fabricated for each composition. Their properties were measured, and then the average values were calculated. The piezoelectric coefficients (*d*_33_) of the piezoelectric ceramics were measured by using a quasi-static piezoelectric *d*_33_ meter (HY2730, Yangzhou, China). The planar electromechanical coupling coefficients (*k*_p_, *k*_31_), piezoelectric coefficient (*d*_31_), mechanical factor (*Q*_m_), dielectric constant (*ε*_r_ = *ε*^T^_33_/*ε*_0_, *ε*_0_ = 8.854 × 10^−12^ F/m), piezoelectric voltage coefficients (*g*_33_, *g*_31_), and Young’s modulus (*Y*^E^_11_) were determined according to the method of resonance and antiresonance frequencies by using an impedance analyzer (HP 4194A, Agilent, Santa Clara, CA, USA) based on the Institute of Electrical and Electronics Engineers (IEEE) standards. Length-mode specimens (22 by 4 by 0.8 mm^3^) were used to calculate *Y*^E^_11_, *k*_31_, *d*_31_, and *g*_31_.

## 3. Results and Discussion

### 3.1. Phase and Microstructure

Figure 2 shows XRD patterns of the 0.55PNN-0.45PZT ceramics doped with different MnO_2_ contents (0, 0.25, 0.5, 1, 2, and 3 mol%). All samples exhibited typical ABO_3_ perovskite structures without the pyrochlore phase. Rhombohedral and tetragonal phases were found to coexist in the PNN-PZT ceramics.

As shown in Figure 3, the XRD patterns were fitted by Gaussian functions in the 2*θ* range of 44.5–45.5°. The (200) peak consisted of three peaks, tetragonal (200) and (002) diffractions peaks presented in green and blue, respectively, and rhombohedral (200) diffraction peak presented in magenta. The red peak represents the overlap intensities of T(200), T(002), and R(200). Figure 2 shows the apparent changes in diffraction peaks, which indicate a gradual rhombohedral-to-tetragonal phase transition. Tetragonality is a crucial structural parameter of the perovskite lattice because it may affect the material properties. The tetragonality was calculated as *I*_T(200)_/*I*_R(200)_, where *I*_T(200)_ is the T(200) intensity and *I*_R(200)_ is the R(200) intensity of the XRD pattern in Figure 3. As shown in Figure 4, with the increase in MnO_2_ content, the tetragonality of the PNN-PZT ceramic initially largely increases, and then saturates at MnO_2_ contents higher than 1 mol% [13].

Figure 5 shows SEM images of the PNN-PZT ceramics with MnO_2_ contents of 0–3 mol%. The pure PNN-PZT ceramic exhibited small grain sizes (microstructures) of almost 1.5 μm. When a small amount of MnO_2_ was added, the grain size of the microstructure was increased. The maximum average grain size of 2.8 μm was observed in PNN-PZT with the MnO_2_ content of 3 mol%.

### 3.2. Dielectric Properties

Figure 6 shows the dielectric constants *ε*_r_ and dielectric losses tan*δ* (%) of the PNN-PZT ceramics with different MnO_2_ concentrations measured at 1 kHz at room temperature. The dielectric loss largely decreased with the MnO_2_ content of 1 mol%, and then steadily increased in the MnO_2_ range of 1 to 3 mol%. The dielectric loss reached the maximum (5.6%) for the undoped PNN-PZT ceramic and minimum (1.6%) for the 1 mol% MnO_2_ sample. Therefore, the dielectric loss decreased by almost five times upon slight MnO_2_ doping. This is consistent with the change rate of the tetragonality. The dielectric loss rapidly decreased with the increase in tetragonality from 0 to 1 mol%. The dielectric loss was improved at MnO_2_ content above 1 mol%, because the tetragonality was saturated at 1 mol%. Figure 6 shows the negative and positive effects on the dielectric constant *ε*_r_ and dielectric loss tan*δ* (%). A negative effect, decrease in dielectric constant, was observed with the increase in tetragonality with the MnO_2_ content. In terms of the dielectric loss, the minimum value could be explained by the competition between the positive effect (increase in tetragonality) and negative effect of the Mn ions on the motion of the wall domains.

With the increase in MnO_2_ content, the dielectric constant decreased because the hardener Mn ions affected the domain movement. The decrease in dielectric constant was caused by the oxygen vacancies generated by the substitutions of the high-valence Ti^4+^ and Zr^4+^ in the perovskite lattice by the low-valence Mn^2+^ and/or Mn^3+^. The dielectric constant rapidly decreased up to the MnO_2_ content of 1 mol%. However, the decrease rate of the dielectric constant was smaller at MnO_2_ contents of 1 to 3 mol%.

### 3.3. Mechanical Properties

Figure 7 shows the mechanical properties of the PNN-PZT ceramics with varying MnO_2_ content (0–3 mol%). The mechanical quality factor *Q*_m_ reflects the steepness of the resonance of the mechanical vibration around the resonance frequency. Therefore, *Q*_m_ has been considered the main parameter of an ultrasonic actuator. *Q*_m_ was improved by approximately five times (from 42.70 to 202.26) with the increase in MnO_2_ content. In addition, the Young’s modulus *Y*^E^_11_ was improved from 7.14 to 10.56 with the increase in MnO_2_ content (a similar curve shape was observed). The changes in mechanical properties with the MnO_2_ content can be attributed to the oxygen vacancies generated by the accepter doping effect. Therefore, the densifications of the ceramics resulted from the introduction of MnO_2_, which improved the piezoelectric properties [13]. The density was improved from 7771.09 to 7938.41 kg/m^3^ with the increase in MnO_2_ content (Table 2). The oxygen vacancies improved the mechanical properties.

### 3.4. Piezoelectric Properties

The electromechanical coupling coefficient (*k*_p_, −*k*_31_) is a constant representing the piezoelectric efficiency of a piezoelectric ceramic, i.e., it represents the efficiency of conversion of electrical energy into mechanical energy. As shown in Figure 8, *k*_p_ decreased with the increase in MnO_2_ content to 0.25 mol%. However, it was largely increased at MnO_2_ contents of 0.25 and 1 mol%, where it reached the maximum of 0.41. Above this concentration, it steadily decreased in the MnO_2_ content range of 1 to 3 mol%.

The curve shape of −*k*_31_ was different. It had the maximum value for the undoped PNN-PZT ceramic, and then steadily decreased with the increase in MnO_2_ content. The −*k*_31_ values at 0.25–1 mol% were similar. At MnO_2_ concentrations less than 1 mol%, the changes may be attributed to the relatively large increase in tetragonality. Above this concentration, *k*_p_ and −*k*_31_ decreased owing to the hardener doping effect (acceptor substitution inducing point defects) [12,13]. The piezoelectric coefficients (−*d*_31_, *d*_33_) reflect the distortion originating from the application of an electric field having a uniform strength without stress. Therefore, the piezoelectric coefficients (−*d*_31_, *d*_33_) have been considered the primary parameters of actuators. The piezoelectric coefficient was calculated by using −*k*_31_, *Y*^E^_11_, and *ε*^T^_33_:
(1)$$d_{3x}=k_{3x}\sqrt{\frac{\varepsilon_{33}^{T}}{Y_{xx}^{E}}}$$

Figure 9 shows the piezoelectric coefficients (−*d*_31_, *d*_33_) of the PNN-PZT ceramics with varying MnO_2_ content (0–3 mol%). −*d*_31_ and *d*_33_ exhibited similar curve shapes; they decreased with the increase in MnO_2_ content because the decrease rate of *ε*^T^_33_ was considerably higher than the increase rates of *k*_31_ and *Y*^E^_11_. The piezoelectric coefficient exhibited the minimum decrease rate between 0.5 and 1 mol% MnO_2_, because of the relatively significant increase in tetragonality at 0.5–1 mol%. 

The piezoelectric voltage coefficients (−*g*_31_, *g*_33_) reflect the field strength, originating from a uniform applied stress without electrical displacement, and thus they represent the sensor properties. The values of −*g*_31_ and *g*_33_ were calculated by using the relationship between the piezoelectric coefficient *d* and dielectric constant, *g* = *d*/*ε*^T^. −*g*_31_ exhibited a similar tendency to that of *k*_31_ (Figure 10). *g*_33_ steadily increased to a high value of 20.31 × 10^−3^ Vm/N at 1 mol%. However, it was decreased at 2 mol%, and then largely increased to the maximum of 21.13 × 10^−3^ Vm/N at 3 mol%, as the decrease rate of *d*_33_ was smaller than the rate of decrease in *ε*^T^_33_. Table 2 shows the densities and dielectric, mechanical, and piezoelectric properties of the PNN-PZT samples with different MnO_2_ contents (0, 0.25, 0.5, 1, 2, and 3 mol%). 

The MnO_2_ doping can affect the PNN-PZT ceramic properties owing to the increase in number of oxygen vacancies generated by the substitutions of the high-valence Ti^4+^ and Zr^4+^ in the perovskite lattice by the low-valence Mn^2+^ and/or Mn^3+^, as mentioned above [4]. The oxygen vacancies improved the mechanical properties (Young’s modulus, *Q*_m_). The Young’s modulus and *Q*_m_ exhibited similar curve shapes. Initially, they were rapidly improved, but their increase rates were reduced at MnO_2_ contents above 1 mol%. The phase of PNN-PZT transformed from rhombohedral to tetragonal with the increase in MnO_2_ content. The phase transition is attributed to the enhancements in electrical properties, which led to the piezoelectric property changes. Therefore, the dielectric constant, Young’s modulus, electromechanical coupling coefficients (*k*_p_,−*k*_31_), and piezoelectric properties (−*d*_31_, *d*_33_, −*g*_31_, and *g*_33_) of PNN-PZT changed with the density of MnO_2_ [4].

## 4. Conclusions

In this study, we investigated the piezoelectric characteristics of 0.55Pb(Ni_1/3_Nb_2/3_)O_3_-0.45Pb(Zr,Ti)O_3_ according to the MnO_2_ additive content (0, 0.25, 0.5, 1, 2, and 3 mol%). We measured *ε*_r_, tan*δ*, *Q*_m_, *Y*^E^_11_, *d*_31_, *d*_33_, *g*_31_, and *g*_33_ as functions of the MnO_2_ concentration. The MnO_2_ addition initially improved the tetragonality of the PNN-PZT ceramic, which then saturated at a MnO_2_ content of 1 mol%.

*Q*_m_ and Young’s modulus (*Y*^E^_11_) also increased with the MnO_2_ content owing to the oxygen vacancies generated by the MnO_2_ doping. The dielectric properties and electromechanical coupling coefficient *k* were optimal at MnO_2_ contents of 0.5–1 mol%, where tan*δ* and *k*_p_ had the maximum values. The changes in electrical properties were attributed to the increased tetragonality. PNN-PZT is a soft piezoelectric material more suitable for actuator applications. The addition of MnO_2_ to PNN-PZT showed its potentials for use in sensory actuators. In following studies, we aim to investigate vibration control and nondestructive testing applications based on the enhanced PNN-PZT ceramics.

## Figures and Tables

**Figure 1 materials-12-04115-f001:**
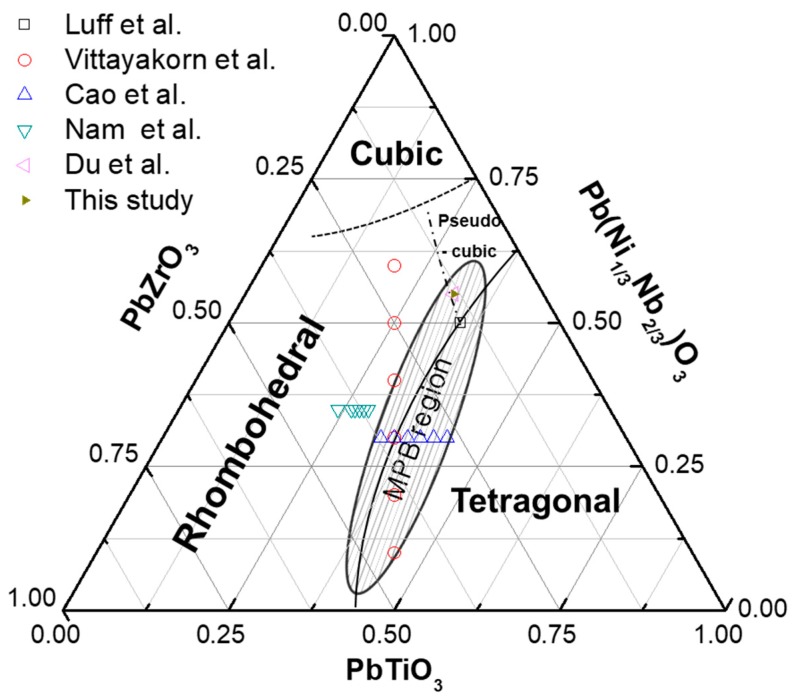
Schematic phase diagram of Pb(Ni_1/3_Nb_2/3_)O_3_-PbTiO_3_-PbZrO_3_ ceramics.

**Figure 2 materials-12-04115-f002:**
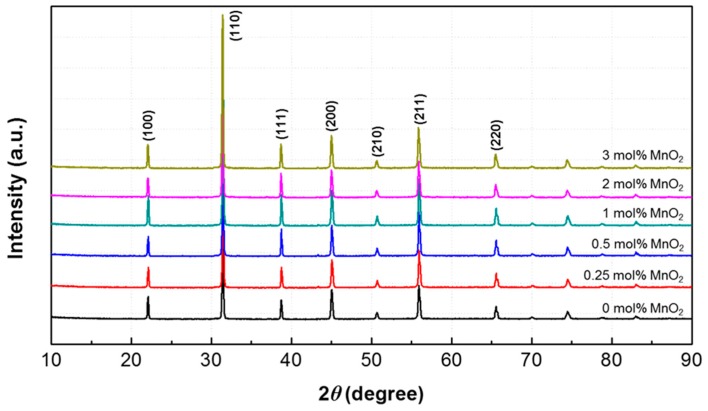
XRD patterns of the 0.55PNN-0.45PZT ceramics with varying MnO_2_ content in the 2*θ* range of 10–90°.

**Figure 3 materials-12-04115-f003:**
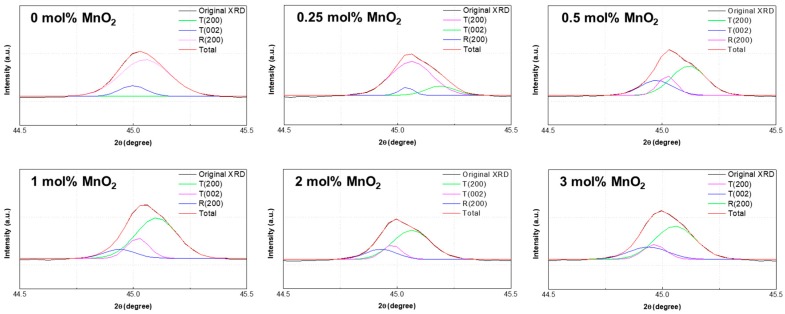
XRD patterns of the 0.55PNN-0.45PZT ceramics with different MnO_2_ contents in the 2*θ* range of 44.5–45.5°. T represents the tetragonal phase, while R represents the rhombohedral phase.

**Figure 4 materials-12-04115-f004:**
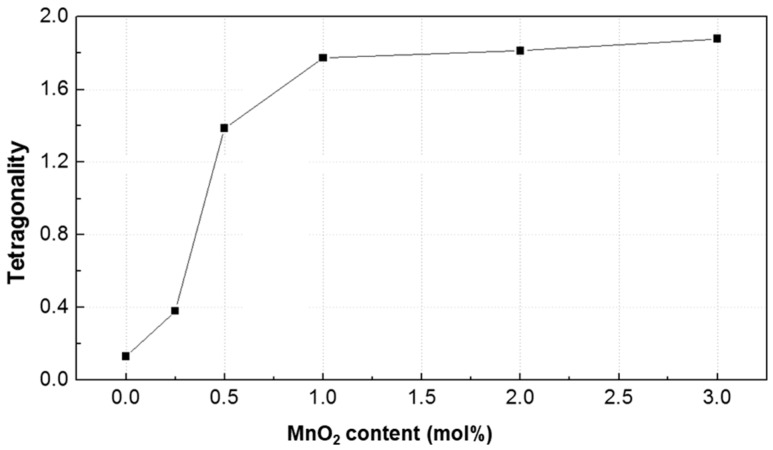
Tetragonality of the 0.55Pb(Ni_1/3_Nb_2/3_)O_3_-0.45Pb(Zr,Ti)O_3_ (PNN-PZT) ceramic as a function of the MnO_2_ content in the range of 0–3 mol%.

**Figure 5 materials-12-04115-f005:**
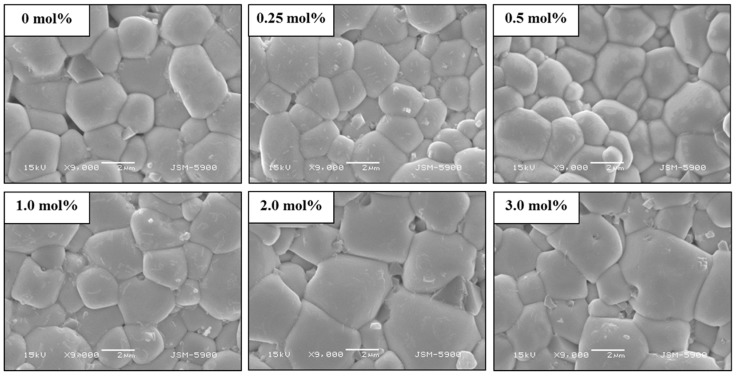
SEM images of the PNN-PZT ceramics with 0–3 mol% MnO_2_ content.

**Figure 6 materials-12-04115-f006:**
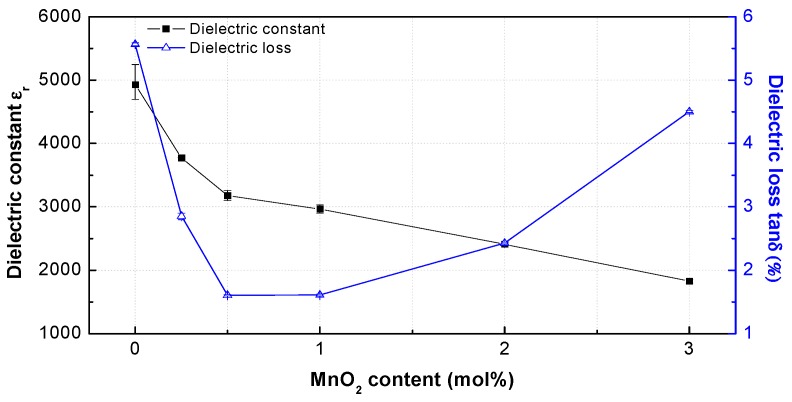
Dielectric constants *ε*_r_ and dielectric losses tan*δ* (%) of the PNN-PZT ceramics with different MnO_2_ concentrations measured at 1 kHz at room temperature.

**Figure 7 materials-12-04115-f007:**
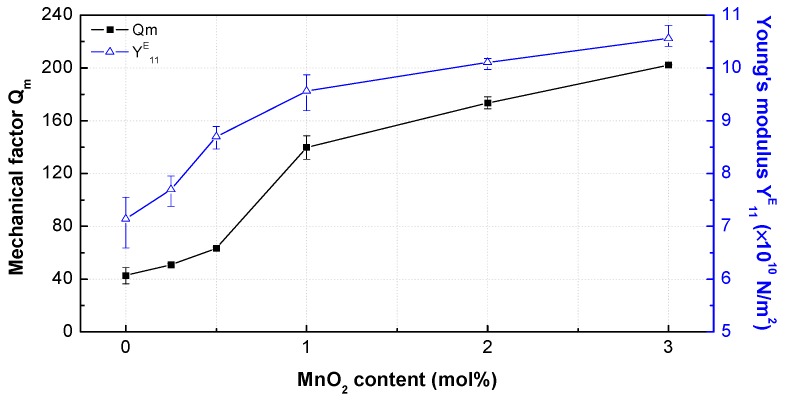
Mechanical factors *Q*_m_ and Young’s moduli *Y*^E^_11_ of the PNN-PZT ceramics with 0–3 mol% MnO_2_ concentrations at 1 kHz.

**Figure 8 materials-12-04115-f008:**
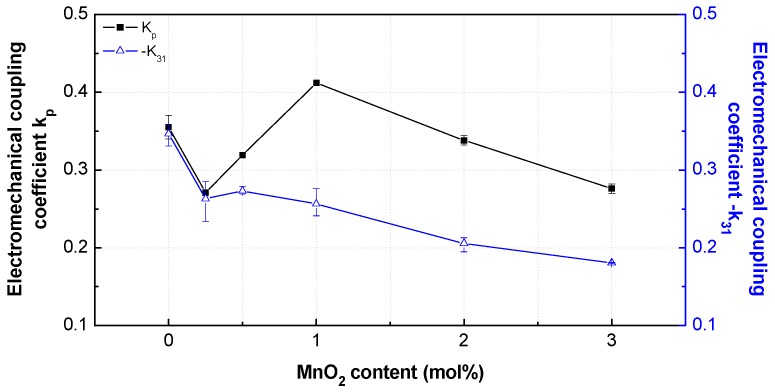
Electromechanical coupling coefficients *k*_p_ and *k*_31_ of the PNN-PZT ceramics with different MnO_2_ concentrations in the range of 0–3 mol%.

**Figure 9 materials-12-04115-f009:**
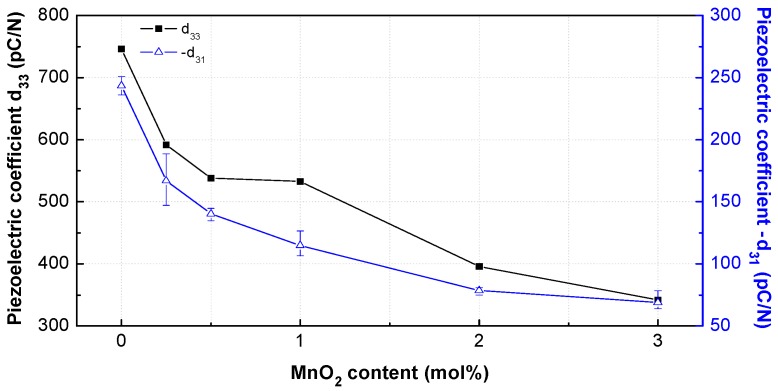
Piezoelectric coefficients *d*_33_ and −*d*_31_ of the PNN-PZT ceramics with different MnO_2_ concentrations in the range of 0–3 mol%.

**Figure 10 materials-12-04115-f010:**
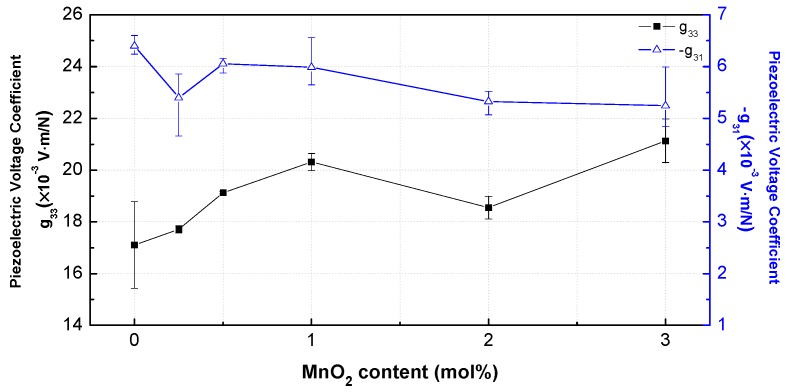
Piezoelectric voltage coefficients *g*_33_ and −*g*_31_ of the PNN-PZT ceramics with different MnO_2_ concentrations in the range of 0–3 mol%.

**Table 1 materials-12-04115-t001:** Literature table of piezoelectric ceramic and properties according to doping materials.

Authors	Piezoelectric Ceramic	Doping	Properties
Yu et al. [9]	0.12Pb(Ni_1/3_Sb_2/3_)-0.48PbTiO_3_-0.40PbZrO_3_	0.15 wt% MnO_2_	*K*_p_ = 0.68, *ε*_r_ = 3069, *Q*_m_ = 181, tan*δ* = 5.4 × 10^−3^
Bamiere et al. [10]	0.674Pb,Nd(Zr, Ti)O_3_-Pb(Ni_1/3_Nb_2/3_)O_3_	SrCO_3_ (0~4 mol%)	Lower the sintering temperature
Du et al. [11]	0.55Pb(Ni_1/3_Nb_2/3_)O_3_-0.45Pb(Zr_0.3_Ti_0.7_)O_3_	Fe_2_O_3_ (0~1.6 mol%)	*ρ* = 7.97 g/cm^3^, *d*_33_ = 956 pC/N, *K*_p_ = 0.74, *ε*_r_ = 6095, tan*δ* = 26 × 10^−3^
Yoo et al. [12]	0.02Pb(Mn_1/3_Nb_2/3_)O_3_-0.12Pb(Ni_1/3_Nb_2/3_)O_3_-0.86Pb(Zr_0.5_Ti_0.5_)O_3_	0.2 wt% MnO_2_	*ρ* = 7.816 g/cm^3^, *d*_33_ = 356 pC/N, *k*_p_ = 0.597, *ε*_r_ = 920, *Q*_m_ = 1186
Liao et al. [13]	0.35BiScO_3_-0.6PbTiO_3_-0.05Pb(Zn_1/3_Nb_2/3_)O_3_	Fe_2_O_3_ (0~1.6 mol%)	Largely reduced tanδ and d_33_, but improved Q_m_
Liu et al. [4]	0.55Pb(Ni_1/3_Nb_2/3_)O_3_-0.45Pb(Zr_0.3_Ti_0.7_)O_3_	1 mol% MnO_2_	*d*_33_ = 710 pC/N, *k*_p_ = 0.595, *ε*_r_ = 3092.25, tan*δ* = 14.9 × 10^−3^, *Q*_m_ = 176

**Table 2 materials-12-04115-t002:** Piezoelectric coefficients *d*_33_ and −*d*_31_ of the PNN-PZT ceramics with different MnO_2_ concentrations in the range of 0–3 mol%.

MnO_2_ Content	Density (kg/m^3^)	*ε*_r_ at 1 kHz	tan*δ* (%) at 1 kHz	*Q* _m_	*Y*^E^_11_ (× 10^10^ N/m^2^)	*K* _31_	*k* _p_	−*d*_31_ (pC/N)	*d*_33_ (pC/N)	−*g*_31_ (× 10^−3^ V∙m/N)	*g*_33_ (× 10^−3^ V∙m/N)
0 mol%	7771.09	4925.51	5.6	42.70	7.14	0.35	0.36	243.40	746	6.40	17.10
0.25 mol%	7782.96	3772.38	2.85	50.99	7.70	0.26	0.27	167.07	592	5.40	17.71
0.5 mol%	7864.18	3177.74	1.6	63.49	8.70	0.27	0.32	140.40	538	6.06	19.13
1 mol%	7936.33	2965.10	1.6	139.70	9.56	0.26	0.41	115.01	533	5.99	20.31
2 mol%	7938.13	2408.93	2.4	173.50	10.11	0.21	0.34	78.51	396	5.33	18.55
3 mol%	7938.41	1826.92	4.5	202.26	10.56	0.18	0.28	68.98	342	5.24	21.13

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
