# Peer review of "Piezoelectric Characteristics of 0.55Pb(Ni_1/3_Nb_2/3_)O_3_-0.45Pb(Zr,Ti)O_3_ Ceramics with Different MnO_2_ Concentrations for Ultrasound Transducer Applications"

_materials, 2019, doi:10.3390/ma12244115_

Round 1
Reviewer 1 Report
English writing requires a serious revision with the help of a native speaker
The literature data reported in the introduction must be given in the table in order to get a better overview.
More generally, the introduction is confuse and does not focus enough on the state of the art of MnO2 additions on PMN-PZT piezoelectric properties and the added value of the author’s paper.
Line 88. The heating ramp used for calcination must be indicated. Did you realize any control of calcined powder (e.g. XRD?). Why 900°C for calcination?
Line 91. Did you perform grains size measurement of your powder?
Line 92. The heating ramp used for binder burning and for the sintering must be indicated
How many samples did you test for each MnO2 mol%?
Figure 3. We can clearly see that the total calculated peak (red) does not exactly fit with the XRD (black) for 0 mol% MnO2. Please indicates the Rwp factors for your fittings. In addition the peak ratio T(002) / T(200) seems to vary strongly with MnO2 mol%, this is not logical. Why don’t you use the Rietveld method (more accurate while taking the whole pattern) to quantify the T and R fractions? In addition Rietvled can shows how varies the parameters a and c for the T phase according to the MnO2 mol% and the crystallite size.
Line 136. How are you grain size calculated?
Lines 145 to 159. Your comments are not clear. I really do not understand how the "changing rate of tetragonality with MnO2 mol%" can influence the properties at a given MnO2 mol%. By comparing fig 4 and 6, dielectric constant seems directly linked to tetragonality (negative effect). For the dielectric loss, the minimum could be explained by the competition between the positive effect of the increase in tetragonality and the negative effect of Mn ions on the motion of the wall domains.
Line 165 to 172. I really do not understand how variations in oxygen vacancies can justify such a strong variation in Young modulus (+ 50% from 7 to 10 1010 N/m2). Unless it is an effect of a better densification?
Did you measure the relative densities of your samples and then the porosity? Control of the effect of MnO2 on the densification can be evidenced from dilatometry curves. Did you check the Young modulus by using another technique (mechanical tests or ultrasouns) ?
How many samples did you test? What does the error bar in your figures represent (several measurements on several specimens?). Why there are no standard deviations in table 1?
Reviewer 2 Report
Lead-based perovskite relaxors offer outstanding piezoelectric properties and have received a great attention. However the potential interdiction of lead-based compositions gives a great impetus on lead-free compositions and this point must be mentioned and some reference given (see the many IEEE papers, e.g. Gouadec et al., IEEE, 2014 JOINT IEEE INTERNATIONAL SYMPOSIUM ON THE APPLICATIONS OF FERROELECTRICS, INTERNATIONAL WORKSHOP ON ACOUSTIC TRANSDUCTION MATERIALS AND DEVICES & WORKSHOP ON PIEZORESPONSE FORCE MICROSCOPY (ISAF/IWATMD/PFM), 63-66; etc.).
The manuscript reports some new data and deserves publication after the following points have been addressed. However the authors failed to highlight the originality of their work and to justify the interest of Mn addition..
In its present status, interest of Fig. 1 is poor. The authors must add unit-cell symmetry and MPB regions, as made for other relaxor composition diagram (see e.g. J. Phys. Chem. Solids 69 (2008) 2503).
Fe and Mn addition has been studied for PMN-PT homologues and it was demonstrated that if Qm increases d33 and e33 decrease. The authors must explain better why they expect positive effect in PNN-PZT compositions.
Lines 68-69 & 224-225: The authors assign the better sinterability of Mn doped perovskites due to an increase of oxygen vacancy number without any references. Add convenient references.
Figure 2 is useless. Magnification is too small to detect all minor phases, if any. Replace with a comparison of 211 and 220 peaks, allowing precise comparison of the distortion of the unit-cells. Are the tetragonality measured with 211 and 220 peaks equal to that measured with 200 one?
Figure 3: enlarge the labels.
Measurement of the densification for each composition is needed, in particular to compare dielectric constant.
Figure 8: the different behavior of Kp and K31 is strange. Explanation not convincing!
As already demonstrated by the literature for perovskite homologues, d33 decreases with Mn addition. This point must be clearly stated and references added.
Table 1 clearly demonstrates that Mn addition decreases the piezo-electric properties, as previously established for homologues.
Miscellaneous
Page 1, line 23: d33?
Lines 60-65 : Comparisons make sense only if the porosity is similar for each sample. Add porosity or densification levels.
Reviewer 3 Report
The manuscript under review devoted to study of piezoelectric properties of PNN-PZT with MnO2 additive. Providing of such investigations is very important for the further application of these materials in actuators and ultrasound transducers. It is understandable due to high macroscopic properties of solid solutions based on PNN-PZT (dielectric, piezoelectric, magnetic, elastic, etc.) and various additives can improve them. Thus, the results presented in the manuscript to determine the correlation between the production method and structure, microstructure, dielectric and piezoelectric properties of solid solutions are actual.
In the manuscript, the author’s presents sufficiently complete information about methods used for study and experimental data. It was shown that all studied samples exhibit a typical ABO3 perovskite structure with rhombohedral and tetragonal phases coexist. The degree of tetragonality varies notably in the concentration range of MnO2 0-1.0. the authors showed a significant effect of the MnO2 additive on the dielectric, piezoelectric, and elastic properties of the studied ceramic solid solutions based on PNN-PZT. For example, dielectric properties decrease almost 5 times, elastic properties increase (mechanical Q factor), and piezoelectric properties decrease (drop of the piezoelectric module from 750 to ~ 320 pC/N). The authors attribute the observed to an increase in the number of oxygen vacancies due to a change in the valence of Mn +2 / +3.
In manuscript all necessary information is captured by 10 figures and 1 table. There are 12 references, all of them are adequate and are reflected in the text.
All studies were conducted at a high scientific level, however the current manuscript not without imperfection:
1) Author’s must check once more Page 5 line 158: “dielectric constant become less as MnO2 content further increase from”. In MnO2, the number 2 must be a subscript;
2) Page 8 line 230: “rhombohedral to tetragonal phase with increasing MnO2 content.”. In MnO2, the number 2 must be a subscript;
3) It is well known that the variable valence of Mn causes the formation of oxygen vacancies in order for the system to maintain electro neutrality. It would be a big plus if the authors added confirmation of this assumption (if it possible).
4) Variation of the piezoelectric characteristics is possible due to a change in the density of the samples. To get the full picture, authors must add the specified data to the manuscript (for example in the table).
However, the obtained results are important both for understanding the physical processes that occur in real objects and for the development of new piezoelectric materials based on PNN-PZT. The described manuscript is sufficient, comprehensive and it corresponds to the field of the Journal «Materials». It should be accepted with minor adjustments.
Round 2
Reviewer 1 Report
Authors have answer to my comments and questions and have greatly improved their paper.
Reviewer 2 Report
All points have been addressed.